# Changes in Sitting Time, Screen Exposure and Physical Activity during COVID-19 Lockdown in South American Adults: A Cross-Sectional Study

**DOI:** 10.3390/ijerph18105239

**Published:** 2021-05-14

**Authors:** Kabir P. Sadarangani, Gabriela F. De Roia, Pablo Lobo, Robinson Chavez, Jacob Meyer, Carlos Cristi-Montero, David Martinez-Gomez, Gerson Ferrari, Felipe B. Schuch, Alejandro Gil-Salmerón, Marco Solmi, Nicola Veronese, Hosam Alzahrani, Igor Grabovac, Cristina M. Caperchione, Mark A. Tully, Lee Smith

**Affiliations:** 1Escuela de Kinesiología, Facultad de Salud y Odontología, Universidad Diego Portales, Santiago 8370179, Chile; 2Department of Kinesiology, Universidad Autónoma de Chile, Santiago 7500912, Chile; 3Laboratorio de Estudios en Actividad Física (LEAF), Universidad de Flores (UFLO), Buenos Aires C1406, Argentina; gabriela.deroia@uflouniversidad.edu.ar (G.F.D.R.); pablo.lobo@uflouniversidad.edu.ar (P.L.); 4Instituto de Salud Pública Andrés Bello, Universidad Andrés Bello, Santiago 8370149, Chile; robinsonchavez@gmail.com; 5Department of Kinesiology, Iowa State University, Ames, IA 50011, USA; jdmeyer3@iastate.edu; 6IRyS Group, Physical Education School, Pontificia Universidad Católica de Valparaíso, Valparaíso 2530388, Chile; carlos.cristi.montero@gmail.com; 7Department of Preventive Medicine and Public Health, Universidad Autónoma de Madrid and IdiPaz, 28049 Madrid, Spain; d.martinez@uam.es; 8Consortium for Biomedical Research in Epidemiology and Public Health (CIBERESP), 28029 Madrid, Spain; 9IMDEA Food Institute, Campus de Excelencia Internacional UAM + CSIC, 28049 Madrid, Spain; 10Escuela de Ciencias de la Actividad Física, el Deporte y la Salud, Universidad de Santiago de Chile (USACH), Santiago 7500618, Chile; gerson.demoraes@usach.cl; 11Department of Sports Methods and Techniques, Federal University of Santa Maria, Santa Maria 97105-900, RS, Brazil; felipe.schuch@ufsm.br; 12International Foundation for Integrated Care (IFIC), Annex Offices, Oxford OX2 6UD, UK; alejandro.gil.salmeron@uv.es; 13Department of Neurosciences, Padua Neurosciences Center, University of Padua, 35122 Padua, Italy; marco.solmi83@gmail.com; 14Geriatrics Section, Department of Medicine (DIMED), University of Padova, 35122 Padova, Italy; ilmannato@gmail.com; 15Institute of Clinical Research and Education in Medicine (IREM), 35122 Padova, Italy; 16Department of Physical Therapy, College of Applied Medical Sciences, Taif University, Taif 21944, Saudi Arabia; halzahrani@tu.edu.sa; 17Department of Social and Preventive Medicine, Centre for Public Health, Medical University of Vienna, 1090 Wien, Austria; igor.grabovac@meduniwien.ac.at; 18Human Performance Research Centre, School of Sport, Exercise and Rehabilitation, University of Technology Sydney, Sydney, NSW 2007, Australia; Cristina.Caperchione@uts.edu.au; 19Institute of Mental Health Sciences, School of Health Sciences, Ulster University, Newtownabbey BT37 0QB, UK; m.tully@ulster.ac.uk; 20The Cambridge Centre for Sport and Exercise Sciences, Anglia Ruskin University, Cambridge CB1 1PT, UK; Lee.Smith@aru.ac.uk

**Keywords:** COVID-19, exercise, sedentary behavior, screen time, public health, health behavior

## Abstract

The worldwide prevalence of insufficient physical activity (PA) and prolonged sedentary behavior (SB) were high before the coronavirus (COVID-19) pandemic. Measures that were taken by governments (such as home confinement) to control the spread of COVID-19 may have affected levels of PA and SB. This cross-sectional study among South American adults during the first months of COVID-19 aims to (i) compare sitting time (ST), screen exposure, moderate PA (MPA), vigorous PA (VPA), and moderate-to-vigorous PA (MVPA) before and during lockdown to sociodemographic correlates and (ii) to assess the impact of lockdown on combinations of groups reporting meeting/not-meeting PA recommendations and engaging/not-engaging excessive ST (≥7 h/day). Bivariate associations, effect sizes, and multivariable linear regressions were used. Adults from Argentina (*n* = 575) and Chile (*n* = 730) completed an online survey with questions regarding demographics, lifestyle factors, and chronic diseases. Mean reductions of 42.7 and 22.0 min./day were shown in MPA and VPA, respectively; while increases of 212.4 and 164.3 min./day were observed in screen and ST, respectively. Those who met PA recommendations and spent <7 h/day of ST experienced greatest changes, reporting greater than 3 h/day higher ST and more than 1.5 h/day lower MVPA. Findings from the present study suggest that efforts to promote PA to South American adults during and after COVID-19 restrictions are needed.

## 1. Introduction

Chronic non-communicable diseases (NCDs), such as cardiovascular and respiratory diseases, type 2 diabetes, and some cancers, are the main cause of morbidity and mortality worldwide [1]. These diseases share four behavioral risk factors including consumption of tobacco and alcohol, unhealthy diet, and insufficient physical activity (PA) [2]. Insufficient PA is the fourth leading cause of death from chronic diseases in both high and middle-income countries [3]. Moreover, 27.5% of the worldwide population is considered to be inactive and, surprisingly, the Latin America and the Caribbean region leads this ranking with 39.1%. Specifically, the prevalence of inactivity among those from Argentina and Chile is 44.2% and 35.1%, respectively [4,5,6]. In order to tackle low levels of PA, measures such as the World Health Organization’s Global Action Plan have been initiated [7]. However, the levels of PA remain low, and social distancing restrictions put in place to reduce transmission during the COVID-19 pandemic may have resulted in further declines in levels of PA.

Prior to COVID-19, the percent of daily time that was dedicated to moderate to vigorous PA (MVPA) in the general population was low (around 1%), in comparison to the time spent in light intensity PA (28%), sleep (29%), and sedentary behavior (42%) [8]. Importantly, high sedentary behavior is also considered to be an independent risk factor concerning NCDs-related morbidity and mortality [9]. Moreover, varying combinations of PA and sedentary behavior may have different influences on mental and physical health. Four different combinations should be considered; when a person falls into the category of insufficient physical activity and excessive time in sedentary behavior is likely the most detrimental to health, whereas the desired combination is when a person is sufficiently active and exhibits low levels of sedentary time. The other two categories include insufficiently active and low levels of sedentary time or sufficiently active and excessive time spent sedentary [10].

Since the beginning of 2020, home confinement as a control measure for the spread of COVID-19, has led to significant changes in daily life. In March 2020, Argentina’s and Chile’s corresponding authorities imposed measures aimed at reducing the transmission of COVID-19 [11,12], influencing people’s lifestyle by implementing isolation guidance in order to slow or prevent the transmission of the virus. Because of this, opportunities for PA have been reduced, and those for sedentary behavior increased across most daily activities (e.g., work and leisure time) [13,14].

To our knowledge, data analyzing the associations between changes in PA and sedentary behavior due to COVID-19 lockdown in South America are scarce. Moreover, Argentina’s and Chile’s contribution to PA research worldwide from 1950 to 2019 represents just 0.31% and 0.73% of all literature, respectively [15,16]. Therefore, the aims of this study in adults from Argentina and Chile during the first months of the COVID-19 pandemic were: (i) to compare sitting time (ST), screen exposure (SE), moderate PA (MPA), vigorous PA (VPA), and MVPA before and during COVID-19 lockdown and the sociodemographic factors associated with them; and, (ii) to assess the impact of lockdown on combination of meeting or not meeting PA recommendations with or without excessive sitting time.

## 2. Materials and Methods

### 2.1. Study Design and Subjects

This cross-sectional study collected information using an online survey in Spanish, during the periods 24 April–27 July 2020 for Argentina (mandatory quarantine was introduced 20 March 2020) and 4 April–26 April 2020 for Chile (mandatory quarantine was introduced 26 March 2020 in some regions). It should be noted that both Argentina and Chile confirmed their first case of COVID-19 on 3 March 2020. Convenience sampling using mass emails to students, colleagues, and researchers’ networks, as well as by social media posts (Twitter, Facebook, Instagram), was used to recruit participants ≥18 years who had been in lockdown for at least seven days and residing in either of these two countries. Lockdown was determined where a participant had not left their home, or only for essential activities (practicing social distancing), either by personal choice, medical indication, or in compliance with mandatory regulations in place. A total of 1483 adults in Argentina and 2520 adults in Chile accepted to participate, with 1305 completing the survey.

The study was conducted according to the guidelines of the Declaration of Helsinki, and was approved by the Universidad de Flores (Argentina) (protocol code 09/2020 and date of approval 22 April 2020) and the faculty of medicine at the Universidad Diego Portales (Chile) (protocol code 02-220 and the date of approval 27 March 2020). Informed consent was obtained from all of the subjects involved in the study.

### 2.2. Power and Sample Size Calculation

The total number of adult populations was around 34,000,000 and 14,500,000, respectively, according to the 2010 Argentinian and 2017 Chilean census. The sample size was calculated by setting the statistical power at a 95% confidence interval with a margin of error of 5% and 4% for Argentina and Chile, respectively. Thus, the required sample size for this study was 384 Argentines and 600 Chileans.

### 2.3. Data Collection

#### 2.3.1. Questionnaire

The participants undertook an approximately 25 min. online questionnaire that examined the impact of lockdown on health and well-being. Information on sociodemographic, social network, COVID-19 knowledge and related restrictions, lifestyle factors (physical and sexual activity, screen exposure, sitting time, tobacco and alcohol consumption), diet, chronic physical diseases, mental health, and wellbeing were gathered.

#### 2.3.2. Sociodemographic Characteristics

Participants’ reported age (18–24, 25–34, 35–44, 45–64 and ≥65 years), sex, area of residence (North, Center, and South or Patagonia for Chile and Argentina, respectively), marital status (single, married/living with a partner, divorces/separated, and widow), and educational attainment (<8 schooling years, 8–12 schooling years, and >12 schooling years).

#### 2.3.3. Chronic Physical Diseases

Self-reported information on current obesity, type 2 diabetes, and hypertension were collected.

#### 2.3.4. Lifestyle Factors

Tobacco and alcohol consumption were reported as yes or no. PA, ST, and SE were all reported as daily hours and minutes prior to and during lockdown. Specifically for PA, daily MPA and VPA were both reported (e.g., before going into lockdown or since going into lockdown, on average how many hours and minutes a day do you spend in: VPA (described as large increases in breathing or heart rate), MPA (described as small increases in breathing or heart rate), SE and ST)), and the participants were categorized as meeting public health recommendations (≥150 min/week of MPA; or ≥75 min/week of VPA; or, an equivalent combination of MVPA throughout the week) [17]. In accordance with the Global Physical Activity Questionnaire [18], participants who reported >960 min/day of PA, were removed from the analysis.

Regarding ST, although there is still no globally agreed cut-off point on the maximum time without substantial deterioration in health [17], there is enough evidence with subjective measurements to suggest that 7+ h a day in ST is likely to have negative effects on health [19], and it was used to categorize participants into high/low ST.

Following the findings that were reported by the Physical Activity Guidelines Advisory Committee [20] of the joint associations of ST and MVPA recommendations with risk of all-cause mortality, the resulting categories of these behaviors were combined into “quadrants” were those cases fulfilling the criteria for both variables (≥150 min/week of MVPA and <7 h of ST) were assigned to the lower-risk quadrant; those failing to fulfill the criteria for both variables were assigned to the higher-risk quadrant, and the two remaining outcomes were considered to be middle-risk quadrants. However, those meeting the PA guidelines were considered to be lower risk in comparison to their inactive counterparts.

### 2.4. Statistical Analysis

Mean, 95% confidence intervals (CI), median, 25th, and 75th percentile were reported for continuous variables, and frequency and percentages for categorical variables. Differences on VPA, MPA, ST, and SE pre-versus during COVID-19 lockdown and, between countries, were analyzed using paired t-test. The effect size (Cohen’s d) was calculated to determine the magnitude of the change, using the following criteria: 0.2 (small), 0.5 (moderate), and 0.8 (large) [21]. Additionally, possible changes in categories of PA recommendations (active or inactive) and ST cut-points (<7 or ≥7 h/day) pre-during lockdown were tested with McNemar’s test.

Multivariate linear regression models were applied estimating the association between differences pre-during COVID-19 lockdown in MPA, VPA, ST, and SE and sociodemographic characteristics (age intervals, sex, educational level, zone of residence, and marital status) for the whole sample and for each country. The results are presented as a standardized beta coefficient allowing for direct comparison across covariates.

The distribution of the sample fulfilling the PA and ST criteria was reported as risk quadrants, both prior and during the lockdown. Longitudinal individual pre-during differences on MVPA and ST were calculated and reported for each quadrant.

All of the statistical analyses were performed using Stata v. 15 (StataCorp LP, College Station, TX, USA). The significance level was set at 5% (*p* < 0.05).

## 3. Results

A total of 1305 adults from Chile and Argentina completed the survey (Argentina, *n* = 575; Chile, *n* = 730). The sample was predominantly composed of females (80.4%), aged between 35 and 44 years (24.3%), with >12 years education (73.3%), and living in central areas (85.4%). When comparing countries, Chile had a younger population, with lower schooling attainment and higher consumption of alcohol and smokers than Argentina (*p* < 0.001). Table 1 details full characteristics of the overall sample and by countries.

Table 2 describes differences in PA, ST, and SE pre-during COVID-19 lockdown. Mean reductions on average of 42.7 (±101.0) min/day in MPA and 22.0 (±66.2) min/day in VPA were shown for the total sample, with major variations in Chile and Argentina, respectively. On the other hand, mean increases in SE of 212.4 (±191.7) min/day and ST of 164.3 (±180.6) min/day were also observed.

Regarding PA recommendations and ST, 56.9% and 34.9% of the sample were classified as active and sitting <7 h during lockdown, respectively.

Appendix A exhibits the differences (pre versus during) of the adjusted associations of VPA, MPA, ST, and SE, with sociodemographic variables, in the total sample and by country. The analysis indicates that there were significant independent associations between ST and SE and age intervals (ST: β −0.13, 95% CI −23.34; −7.24, SE: β −0.16, 95% CI −28.88; −11.93), and women (ST: β 0.06, 95% CI 2.93; 51.77, SE: β 0.07, 95% CI 7.87; 59.46). Significant independent associations were found between VPA and female sex (β 0.06, 95% CI 0.56; 18.84) and marital status (β 0.08, 95% CI 1.26; 12.69) with limited differences in the effect of sociodemographic variables between countries.

Figure 1 represents risk-quadrants according to PA recommendations and ST cut-point. 9.1% of the total sample was classified in the higher-risk quadrant and 54.3% in the lower-risk quadrant previous to COVID-19 pandemic. However, during lockdown, the number of cases tripled in the higher risk quadrant and halved in the lower risk quadrant. Regarding country-specific quadrants, Chile’s higher-risk and Argentina’s lower-risk scenario worsened from 8.9% to 31.5% and from 53.0% to 18.6%, respectively.

Table 3 displays longitudinal changes according to risk quadrants. The lower-risk quadrant experienced poorer results, increasing in more than 3 h and decreasing in more than 1.5 h both ST and MVPA, respectively. Moreover, Argentina experienced a higher overall increase in ST, while Chile suffered the largest decline in MVPA.

Appendix A represents the changes of participants who were active/inactive and spending/not-spending excessive ST. Among 1029 active participants, 346 became inactive during the lockdown. On the other hand, among 276 inactive adults, 60 became active during the lockdown. Regarding ST, among the 865 participants who spent <7 h/day in this behavior, 471 increased their levels during lockdown.

## 4. Discussion

This cross-sectional study compared ST, SE, MPA, VPA, and MVPA before and during lockdowns on sociodemographic correlates and assessed the impact of COVID-19 on quadrants combining PA recommendations and ST among South Americans adults. One of the main findings of this paper is the reduction of MPA and VPA min./day during lockdowns, lowering the percentage of active participants by 22.0% (*n* = 286). Secondly, an increase of 3.5 h in SE and 2.7 h/day in ST led to an enrichment in sedentary behaviors. Furthermore, it was also shown that participants who were previously classified in the lower-risk quadrant, engaging in ≥150 min/week of MVPA and sitting <7 h, suffered major negative effects with the lockdowns.

In the present study, a small effect size with a significant reduction of 38% in MPA and 50% in VPA was observed, which is consistent with other research [13,22,23]. Moreover, in the study that was conducted by Ammar et al., (2020), including 35 research organizations from different continents, mean differences of 33.4% and 33.1% for MPA and VPA, were observed, respectively [22]. However, their study had a higher proportion of males (46.2%), higher educational attainment (90.9%), and lower PA levels pre-isolation, thus allowing for smaller differences between periods. Despite our findings, one study found 30 min/day differences in MPA only [24], while, another, 37 min/day variation in VPA exclusively [25]. These differences in MPA, but not in VPA, could be attributed to their sample inclusion criteria, as all participants declared having chronic conditions, experiencing greater limitations for VPA [24]. On the other hand, VPA reductions, per se, could be due to the closure of sport facilities or restrictions in green spaces, whereas the levels of MPA may had been sustained with alternative activities at home, including PA online classes [13,25].

Sedentary behaviors, such as ST and SE, increased significantly (ST: d = 0.91; SE d = 1.11), especially in women and in young adults, being consistent with other studies from the United States and Spain [25,26]. This increase among those aged 18–24 years during lockdowns could be explained by university campus shutdowns shifting to online classes summed with restrictions on movement and social life. Focusing on the differences of ST, our study found a 73.30% increase during isolation, percentages that are comparable with other studies [22,25] but much higher than one study observed that was carried out in Scotland [27], which only found a 29.1 min./day mean difference. The differences between studies could be due to a younger population sample (50+: 27.8% in the present report vs. 45.8%) and lower ST pre-lockdown in the present study.

Pre-lockdowns findings suggest that eight out of 10 participants reported meeting PA recommendations, which is consistent with a study from the United States [28], but higher than one that was conducted in Italy [29], which found a 55.5% prevalence. However, when analyzing changes during lockdown, our study found a 26.5% reduction in PA levels, in comparison to the 39.0% and 63.0% in the aforementioned studies. This lower reduction in the present study could be due to the overrepresentation of women being inactive at baseline, and not including walking as PA, clearly reducing this behavior with the “stay at home” advice. On the other hand, 4.6% of inactive participants became active, figures that are significantly lower than other studies that found a 25.2% and 22.0% increase [28,29]. Additionally, changes among active participants during the lockdown are higher in 15.6% than that observed in Italy [29].

Current research on the interaction between PA and sedentary behavior is that high levels of PA have a protective effect over the adverse health effects of sedentary behavior [10,20]. Regarding the distribution of the sample across quadrants prior and during COVID-19 isolation (transversal changes), the greatest changes in percentages of participants were observed in the extreme quadrants (lower-risk from 54.3% to 22.8%, and higher-risk from 9.1% to 30.9%). For case-by-case changes in behavior (longitudinal modification), the results of the present study show a significant increase in ST of the entire sample, regardless of their habits before isolation. However, this increase is even more pronounced in the lower-risk quadrant; they suffered the greatest decrease in PA and, consequently, lost the protective effect that this would have on prolonged sitting, thus increasing the risk of all-cause mortality [9].

Despite efforts to keep people physically active during social isolation [30], the implemented strategies do not appear to have been effective in increasing PA and reducing ST [31]. This situation has an immediate impact on people’s health because of the negative effect on the immune system and, thus, a greater predisposition to contagion [32]. Two recent studies have recently shown that high levels of physical activity produce an immune protective effect by attenuating the symptoms of SARS-CoV-2 and, thus, the subsequent prognosis [33,34]. In “at-risk” individuals, i.e., those with pre-existing comorbidities prior to virus infection (i.e., individuals with obesity or overweight, insulin resistance, and diabetes) have had worse symptoms during COVID-19 infection and a worse subsequent prognosis. Especially patients with obesity seem to have less success post-hospitalization, as increased adiposity may damage the pulmonary microenvironment with circulating inflammatory cytokines and, thus, contribute to an increased cycle of local inflammation and secondary injury [35].

This situation continues to cause widespread personal suffering, along with severe pressure on medical and health care providers. Moreover, these results have implications for direct and indirect costs in health care systems. Ding et al., (2016) observed that insufficient PA costs $53.8 billion globally in 2013 [36]. Moreover, deaths that are attributable to physical inactivity adds another $13.7 billion globally in productivity losses annually and results in 13.4 million disability-adjusted life-years globally. Estimates from low and middle-income countries (LMIC) and high-income countries indicate that between 2–3% of national health care expenditures are attributable to physical inactivity [37]. In addition, ST also appears costly, as Heron et al., (2019) estimated that prolonged ST (≥6 h) costs the United Kingdom £0.8 billion annually [38].

The socio-cultural changes that have taken place concerning insufficient PA and sedentary behavior during the COVID-19 period, referred to as the “double pandemic” [39] are uncertain, and these changes are determinant in the way people work, move around, and use their free time. Therefore, it is likely that, when the COVID-19 pandemic ends, the issues of insufficient PA and sedentary behavior need to be reinforced and addressed urgently, avoiding the likely increase in NCDs morbidity and mortality, thus healthcare system costs. Therefore, it is necessary to design a regional action plan to promote PA and reduce sedentary behavior to face the uncertain “new normal”.

### Limitations and Strengths

The limitations of this study include a non-representative sample from the involved countries limiting generalization. Self-reported and recall of pre-isolation activity can incur bias into the findings. However, isolated questions on PA and ST were derived from the global physical activity questionnaire instrument, which is validated in both countries [40]. Additionally, the cross-sectional design is incapable of establishing causality.

On the other hand, strengths include a survey that has been used in many countries (United Kingdom, Spain, Italy, Jordan, Brazil, United States, and Australia). Moreover, studies on changes in PA, ST, and SE due to COVID-19 isolation or combining these behaviors are inexistent in South American adults.

## 5. Conclusions

This study is the first in South America indicating the detrimental effects of COVID-19 isolation on MPA, VPA, ST, and SE in adults. The promotion of PA and reducing sedentary behaviors with public strategies are needed in order to mitigate the behavioral effects of COVID-19 isolation. Shifts into high-risk categories (low PA and high ST) that have the potential to extend beyond lockdown are leaving a population at risk of major medical concerns needing to be supported by a healthcare system that is weakened by the pandemic itself. Therefore, a major challenge lies ahead: investment is needed to promote physical activity as part of the plan to remedy the consequences of COVID 19. To better understand the long-term consequences, prospective follow-up studies comparing the countries involved and various containment and health promotion strategies are needed, in order to provide robust information for decision makers during and after pandemics.

## Figures and Tables

**Figure 1 ijerph-18-05239-f001:**
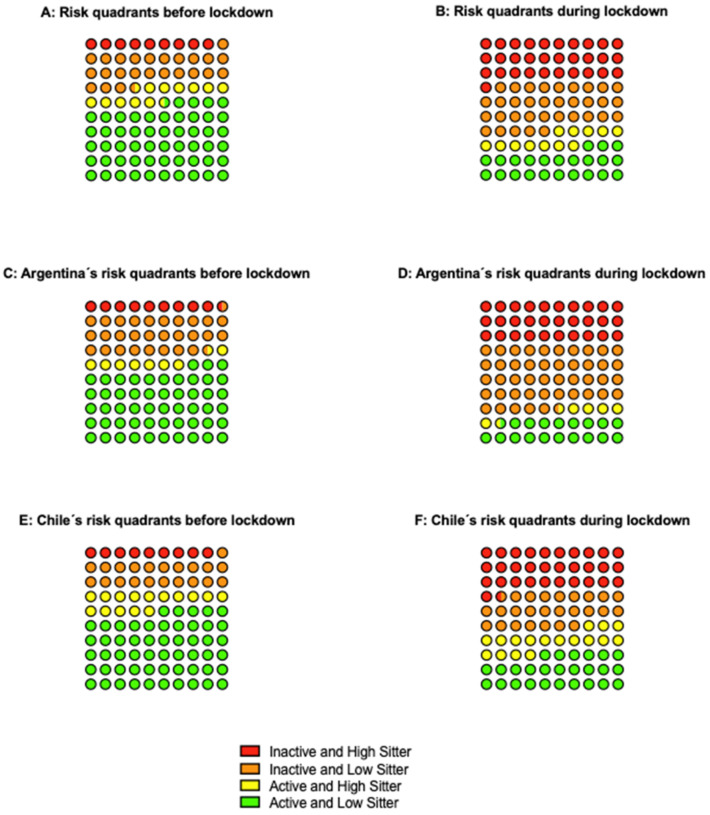
Risk quadrants according to physical activity recommendations and sitting time.

**Table 1 ijerph-18-05239-t001:** Descriptive characteristics of the overall sample and by country.

Variable		Total (*n* = 1305)	Argentina (*n* = 575)	Chile (*n* = 730)	*p*
		*n* (%)	*n* (%)	*n* (%)	
Age intervals	18–24	157 (12.0%)	36 (6.3%)	121 (16.6%)	<0.001
	25–34	239 (18.3%)	111 (19.3%)	128 (17.5%)	
	35–44	317 (24.3%)	128 (22.3%)	189 (25.9%)	
	45–54	229 (17.5%)	95 (16.5%)	134 (18.4%)	
	55–64	238 (18.2%)	121 (21.0%)	117 (16.0%)	
	≥65	125 (9.6%)	84 (14.6%)	41 (5.5%)	
Gender	Male	256 (19.6%)	119 (20.7%)	137 (18.8%)	0.384
	Female	1049 (80.4%)	456 (79.3%)	593 (81.2%)	
Area of Residence	NOA + NEA/North	62 (4.8%)	26 (4.5%)	36 (4.9%)	0.112
	Center/Center	1114 (85.4%)	481 (83.7%)	633 (86.7%)	
	Patagonian/South	129 (9.9%)	68 (11.8%)	61 (8.4%)	
Marital Status	Single	460 (35.2%)	165 (28.7%)	295 (40.4%)	<0.001
	Married/Living with Partner	593 (45.4%)	286 (49.7%)	307 (42.1%)	
	Divorced/Separated	213 (16.3%)	94 (16.3%)	119 (16.3%)	
	Widowed	39 (3.0%)	30 (5.2%)	9 (1.2%)	
Educational Attainment	Elementary School/<8 years	63 (4.8%)	15 (2.6%)	48 (6.6%)	<0.001
	High School/8–12 years	286 (21.9%)	147 (25.6%)	139 (19.0%)	
	Tertiary or University/>12 years	956 (73.3%)	413 (71.8%)	543 (74.4%)	
Smoking/Tobacco Use	Yes	319 (24.4%)	95 (16.5%)	224 (30.7%)	<0.001
Alcohol Consumption	Yes	552 (43.7%)	221 (38.4%)	331 (48.0%) ^a^	0.001
Obesity	Yes	230 (21.3%)	115 (20.0%)	115 (22.7%) ^b^	0.282
Hypertension	Yes	189 (17.5%)	102 (17.7%)	87 (17.2%) ^b^	0.802
Type 2 Diabetes	Yes	54 (5.0%)	24 (4.2%)	30 (5.9%) ^b^	0.189

^a^*n* = 689; ^b^
*n* = 507.

**Table 2 ijerph-18-05239-t002:** Changes in physical activity, sitting time, and screen time.

Variable	Before COVID-19 Lockdown	During COVID-19 Lockdown	Difference	Cohen’s d	*p*
	Mean (95% CI ^a^)	Mean (95% CI ^a^)	Delta %		
Moderate PA ^b^ (min/day)					
Total	99.0 (93.0; 105.0)	56.3 (51.1; 61.5)	−37.30%	−0.42	<0.001
Argentina	92.9 (85.2; 100.7)	55.7 (48.6; 62.8)	−32.60%	−0.39	<0.001
Chile	103.7 (95.0; 112.5)	56.7 (49.3; 64.2)	−41.40%	−0.45	<0.001
Vigorous PA ^b^ (min/day)					
Total	44.1 (39.4; 48.7)	22.1 (18.9; 25.2)	−49.50%	−0.33	<0.001
Argentina	36.6 (31.0; 42.2)	16.2 (13.1; 19.3)	−54.50%	−0.35	<0.001
Chile	49.9 (43.0; 56.9)	26.7 (21.6; 31.8)	−45.60%	−0.32	<0.001
Sitting Time (min/day)					
Total	341.8 (332.7; 350.9)	506.1 (494.4; 517.9)	73.30%	0.91	<0.001
Argentina	377.2 (363.0; 391.4)	549.6 (533.0; 566.2)	71.20%	0.99	<0.001
Chile	314.0 (302.5; 325.4)	471.9 (455.9; 487.9)	74.90%	0.85	<0.001
Screen Time (min/day)					
Total	258.0 (248.8; 267.2)	470.4 (457.8; 483.0)	130.60%	1.11	<0.001
Argentina	272.6 (258.1; 287.0)	489.2 (470.9; 507.4)	129.10%	1.17	<0.001
Chile	246.5 (234.6; 258.4)	455.6 (438.3; 473.0)	131.70%	1.07	<0.001
	**Before COVID-19 Lockdown**		**During COVID-19 Lockdown**		
	***n* (%)**	**95% CI**	***n* (%)**	**95% CI ^a^**	
PA ^b^ ≥ 150 min/week					
Total	1029 (78.9%)	76.6–81.1%	743 (56.9%)	54.2–59.6%	<0.001
Argentina	473 (82.3%)	79.1–85.4%	367 (63.8%)	59.9–67.8%	<0.001
Chile	556 (76.2%)	72.9–79.2%	376 (51.5%)	47.8–55.2%	<0.001
ST ^c^ < 7 h					
Total	865 (66.3%)	63.7–68.8%	456 (34.9%)	32.4–37.5%	<0.001
Argentina	353 (61.4%)	57.4–65.4%	142 (24.7%)	21.2–28.2%	<0.001
Chile	512 (70.1%)	66.7–73.4%	314 (43.0%)	39.4–46.7%	<0.001

^a^ Confidence interval, ^b^ Physical activity, ^c^ Sitting time.

**Table 3 ijerph-18-05239-t003:** Changes in sitting time and moderate to vigorous physical activity according to pre-lockdown risk quadrants.

Pre-Lockdown Risk Quadrant	Δ Sitting Time (min/day)		Δ MVPA ^a^ (min/day)	
	Median (*p* 25%; *p* 75%)	Mean (95% CI ^b^)	Median (*p* 25%; *p* 75%)	Mean (95% CI ^b^)
**Lower Risk: Active and Low Sitting**				
Total (*n* = 708)	180.0 (120.0; 300.0)	196.2 (184.3; 208.0)	−60.0 (−150.0; −18.8)	−97.0 (−107.7; −86.2)
Argentina (*n* = 305)	180.0 (120.0; 300.0)	210.8 (194.6; 226.9)	−60.0 (−140.0; −20.0)	−84.8 (−98.4; −71.2)
Chile (*n* = 403)	180.0 (80.0; 300.0)	185.1 (168.2; 202.0)	−90.0 (−240.0; −30.0)	−143.5 (−166.2; −120.8)
**Medium Risk: Active and High Sitting**				
Total (*n* = 321)	120.0 (0.0; 240.0)	112.2 (91.2; 133.1)	-60.0 (−120.0; 0.0)	−69.3 (−83.5; −55.1)
Argentina (*n* = 168)	120.0 (0.0; 240.0)	116.4 (88.5; 144.2)	−45.0 (−90.0; 0.0)	−64.0 (−83.2; −44.7)
Chile (*n* = 153)	120.0 (0.0; 240.0)	107.5 (75.6; 139.4)	−60.0 (−180.0; 0.0)	−92.4 (−123.0; −61.8)
**Medium Risk: Inactive and Low Sitting**				
Total (*n* = 157)	150.0 (110.0; 240.0)	181.3 (154.4; 208.2)	0.0 (0.0; 0.0)	19.0 (6.2; 31.8)
Argentina (*n* = 48)	240.0 (120.0; 337.5)	218.2 (166.0; 270.4)	0.0 (0.0; 3.8)	16.8 (5.9; 27.6)
Chile (*n* = 109)	120.0 (90.0; 240.0)	165.1 (133.9; 196.3)	0.0 (0.0; 0.0)	23.3 (4.1; 42.5)
**Higher Risk: Inactive and High Sitting**				
Total (*n* = 119)	120.0 (0.0; 225.0)	93.0 (54.1; 132.0)	0.0 (0.0; 30.0)	29.1 (12.8; 45.5)
Argentina (*n* = 54)	120.0 (0.0; 240.0)	89.4 (36.7; 142.0)	5.0 (0.0; 60.0)	48.8 (14.7; 82.9)
Chile (*n* = 65)	90.0 (0.0; 180.0)	96.1 (38.4; 153.8)	0.0 (0.0; 0.0)	18.4 (6.0; 30.7)

^a^ Moderate-to-vigorous physical activity. ^b^ Confidence interval.

## Data Availability

Not applicable.

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
