# Peer review of "Changes in Sitting Time, Screen Exposure and Physical Activity during COVID-19 Lockdown in South American Adults: A Cross-Sectional Study"

_ijerph, 2021, doi:10.3390/ijerph18105239_

Round 1
Reviewer 1 Report
Very interesting research.
Good review of and approach to the problem, and a faiorly adequate use of references, though I miss some, published in this very journal, that could have helped to stablish valuable comparisons. E.g.:
Salvador Angosto, Rosendo Berengüí, José Miguel Vegara-Ferri and José María López-Gullón. Motives and Commitment to Sport in Amateurs during Confinement: A Segmentation Study. Int. J. Environ. Res. Public Health 2020, 17(20), 7398; https://doi.org/10.3390/ijerph17207398
or
Rosendo Berengüí, José María López-Gullón and Salvador Angosto. Physical Sports Activities and Exercise Addiction during Lockdown in the Spanish Population Int. J. Environ. Res. Public Health 2021, 18(6), 3119; https://doi.org/10.3390/ijerph18063119
Another minor point I would like to highlight is the advisability of, apart from the lines devoted to the study design, adding a brief description of the statistical analysis in the abstract.
Results and conclusions seem to complement previous related findings on the matter. The methodology, data analysis and results sections all were good as well as the discussion section.
The manuscript deals with a relevant topic, i.e., the commitment to sport, sports habits and the profile of two countries’ populations (Chile and Argentina) during lockdown. I think it is a positive addition to the analysis of circumstances that would allow for the design of strategies intended to mitigate the effects of Covid-19 through physical activity. The present study undoubtedly adds insight to what is known about the subject matter and opens up new perspectives for future research.
Reviewer 2 Report
Dear authors and editor,
The manuscript titled "Changes in sitting time, screen exposure and physical activity during Covid-19 lockdown in South American adults: A cross-sectional study" This is a cross-sectional descriptive study that compare sitting time (ST), screen exposure, moderate PA (MPA), vigorous PA
(VPA) and moderate-to-vigorous PA (MVPA) before and during lockdown to sociodemographic correlates and to assess the impact of lockdown on combinations of groups reporting meeting/not-meeting PA recommendations and engaging/not-engaging excessive ST (≥7 hrs/day)
The manuscript has several biases that cause the results to be assessed with some caution. However, the authors are aware of these limitations.
There are many minor and major issues I'd like the authors resolve.
Abstract
1-Add the study design to the abstract.
2- Change the keywords. Delete the words "sedentary lifestyle". Not found in the MeSH (Medical Subject Headings). Change to "Sedentary Behavior".
Introduction
3-It is recommended that this section be expanded.
Materials and Methods
4-Study size: Explain how the study size was arrived at.
Measuring instruments have certain limitations. Sport activity has a unit of measurement (METs). METs are used to compare the energy cost of different activities. Sports activity is not only determined by the number of hours per week. The intensity influences whether the physical activity is moderate or vigorous.
Statistical Analysis: Median and 25th-75th percentiles are used when samples are nonparametric. The authors use the mean together with the median in the descriptive analysis.
Results
- adequate
Discussion
- adequate
Conclusion
- adequate
Reference:
- adequate
Author Response
"Please see the attachment."

Round 2
Reviewer 2 Report
I am satified about revised version. The authors were very responsive to the concerns expressed in the initial review.